# Genome-Scale Reconstruction of Microbial Dynamic Phenotype: Successes and Challenges

**DOI:** 10.3390/microorganisms9112352

**Published:** 2021-11-14

**Authors:** Nicolai S. Panikov

**Affiliations:** Department of Chemistry and Chemical Biology, Northeastern University, 360 Huntington Ave., Boston, MA 02115, USA; n.panikov@northeastern.edu

**Keywords:** growth kinetics, survival, death, substrate limitation, starvation, gene expression, conditional expression of macromolecules, protein allocation, batch culture, chemostat, cell cycle, metabolic network, pool, metabolic intermediates, cell composition, biomass brutto-formulae, kinetic order

## Abstract

This review is a part of the SI ‘Genome-Scale Modeling of Microorganisms in the Real World’. The goal of GEM is the accurate prediction of the phenotype from its respective genotype under specified environmental conditions. This review focuses on the dynamic phenotype; prediction of the real-life behaviors of microorganisms, such as cell proliferation, dormancy, and mortality; balanced and unbalanced growth; steady-state and transient processes; primary and secondary metabolism; stress responses; etc. Constraint-based metabolic reconstructions were successfully started two decades ago as FBA, followed by more advanced models, but this review starts from the earlier nongenomic predecessors to show that some GEMs inherited the outdated biokinetic frameworks compromising their performances. The most essential deficiencies are: (i) an inadequate account of environmental conditions, such as various degrees of nutrients limitation and other factors shaping phenotypes; (ii) a failure to simulate the adaptive changes of MMCC (MacroMolecular Cell Composition) in response to the fluctuating environment; (iii) the misinterpretation of the SGR (Specific Growth Rate) as either a fixed constant parameter of the model or independent factor affecting the conditional expression of macromolecules; (iv) neglecting stress resistance as an important objective function; and (v) inefficient experimental verification of GEM against simple growth (constant MMCC and SGR) data. Finally, we propose several ways to improve GEMs, such as replacing the outdated Monod equation with the SCM (Synthetic Chemostat Model) that establishes the quantitative relationships between primary and secondary metabolism, growth rate and stress resistance, process kinetics, and cell composition.

## 1. Introduction


*Everything should be made as simple as possible... but not simpler.*
Albert Einstein

Presently, the NCBI lists more than 300,000 completed whole-genome projects. With ~7500 sequenced fungal species and more than 31,000 prokaryotic (archaeal and bacterial) OTUs, we can state that whole-genome sequence data are now available for nearly all cultivable microorganisms of medical, industrial, or environmental significance. Whole-genome projects are expected to increase over time [1] to further expand their coverage and as strain-specific resequencing, metagenomic surveys of communities, etc. Yet, let us face it, the main bottleneck now is not the number of sequenced genomes but, rather, our limited capacity to capitalize on the already accumulated genomic information [2]. Consequently, modern bioscience is confronted with the challenge to translate sequence data into solid biological knowledge, better industrial and healthcare solutions, and an improved mechanistic understanding of bioprocesses. This challenge can be met by system biology [3], using as research tools genome-scale metabolic reconstructions and genome-scale models (GEMs). The ambitious goal of GEM is the accurate prediction of the phenotype from the respective genotype under specified environmental conditions. This undoubtedly would be a remarkable scientific breakthrough, bringing formerly descriptive biology to the top of natural sciences.

The most popular version of GEM, called FBA (Flux Balance Analyses), has been already established as a powerful tool in metabolic engineering and diverse applications [1,4,5,6,7,8,9,10,11,12]. It uses steady-state approximation and other simplifying assumptions that greatly improve the speed of computations with inevitable sacrifices, one of them being the static way of presenting metabolic reconstructions that reproduce a screenshot of cellular metabolism beyond the time axes. More advanced contemporary GEMs, such as dFBA, ME, and WC models, have been charged with the goal of the dynamic reconstruction of metabolism based on the simulation of natural regulatory mechanisms. The correct understanding of these mechanisms should allow the adequate reconstruction of biodynamic behaviors of microorganisms, from trivial binary division to a real-life microbial dynamic under specified environmental conditions: the balanced and unbalanced growth, steady-state and transient processes, survival and recovery from stresses, cell differentiation, biosynthesis of products, including secondary metabolites, etc. Taken together, the listed processes represent a dynamic phenotype [13] of microorganisms. However, modern dynamic GEMs are applied nearly exclusively to trivial data, such as exponential growth with constant SGR. It is well-justified at the initial stages of the methodology development; there is no doubt that new computational approaches should be tested against as simple of data as possible.

Today, genomic-scale system biology is mature enough to address the famous A. Einstein maxim (see epigraph) that gives sagacious advice to carefully balance between the simple and oversimplified. Too simple of data can hardly inspire generation of strong hypotheses crucial for development of systems biology, restrain practical GEM applications, and generally discourage research progress. This review illustrates typical nontrivial microbial behaviors that are commonly observed in laboratory experiments, industrial bioreactors, and in nature. The aim of this review is to show that complex biodynamics is not a confusing mixture of chaotic events but, rather, a highly reproducible phenomena that can be accurately recorded and interpreted using mechanistic mathematical models.

The review starts by defining the key terms and basic principles of growth theory to explain the distinction between simple and complex biodynamics of microorganisms. Next, we review the nongenomic predecessors of the currently available GEMs to show that some of them are built upon the outdated frameworks compromising their performances. Luckily, such deficiencies can be fixed in a straightforward way. Finally, we discuss the prospects for improvements of GEMs for the purpose of dynamic simulations. The technical details, including mathematical equations, their interpretation, alternative solutions, etc., the interested reader can find in the Appendix B.

## 2. Simple and Complex Growth: The Basic Principles

Here, we are going to justify the following basic principles used throughout the review:The microbial specific growth rate (SGR) depends on environmental conditions and the macromolecular cell composition (MMCC). MMCC variation is the result of self-regulation, including differential gene expression in response to a changeable environment.Variable MMCC is the main reason for the higher complexity of microbial growth kinetics vs. chemical and enzymatic reactions.A nutrient supply is the primary environmental factor controlling microbial behavior; the effects of other modifying factors (T, pH, osmolarity, etc.) can be understood only in combination with the nutritional factor.MMCC dependence on SGR is a common misconception; both complex variables depend on the environmental conditions.A growth is called *simple* (syn: balanced and steady state) if it proceeds exponentially under steady environmental conditions with constant SGR and MMCC. Otherwise, the cell dynamics are *complex*.

### 2.1. MMCC and Growth Kinetics

Microbial growth under various tested conditions is usually recorded as a time series of cell mass concentration (*x*), nutrient substrates (*s*), and products (*p*). A specific growth rate (SGR) found by the numeric differentiation of growth curves (corrected for elimination in open systems, e.g., the dilution rate in a chemostat) is the most inclusive biokinetic variable. There are well-defined mass–balance relationships between SGR and other growth characteristics, e.g., the apparent yield, maintenance, respiration, substrate uptake, and products formation [14,15]. The listed biokinetic terms describe growth in a reductionistic ‘chemical’ way, but there are also remarkable changes in the quality of cells, e.g., size and shape, activities, stress resistance, productivity, etc., that, in the past, were combined under the vague term physiological state. Presently, the state of growing cells is surveyed by using omics molecular inventories on top of conventional chemical analyses of cells (elemental composition, total proteins, nucleic acids, lipids, etc.). About 10% of the cell dry mass is accounted for as a pool of small molecules using metabolomic analyses, and ~90% is represented by macromolecules evaluated by proteomic, glycomic, and lipidomic profiling. This review focuses on the macromolecular cell composition (MMCC) as a principal effector of growth kinetics; the metabolic pool is touched only from a biokinetic perspective (Section B.2.3).

Historically, the first growth-associated changes in MMCC were established ~60 years ago for the most abundant cellular constituent, the ribosomes making up to 50% of the total cell mass in fast-growing cells and dropping down to 5–10% under growth deceleration [16,17]. At the same time, the opposite trend was found for the storage polymers (glycogen in *E. coli*), which progressively decreased with the growth acceleration from 23 to 2% [18]. Presently, the changes in MMCC have been mostly followed by transcriptomic and proteomic techniques being referred to as ‘condition-specific expression’ [19,20,21]. A commonly accepted view is that the up- and downregulation of proteins is one of the most essential and fundamental qualities of life, the result of metabolic regulation and differential gene expression in response to environmental stimuli. However, this quality turned out to be difficult to incorporate into mechanistic growth models (see below).

### 2.2. Limited Applicability of Enzyme Kinetics to Microbial Growth

Enzymes are simpler than cells, and enzyme kinetics has a longer history of successful developments. It was the main reason for numerous attempts to adapt enzymological models to microbial growth. The obvious distinction between enzymes and growing microorganisms is that enzymes do not proliferate over time. However, there is a simple way to handle this complication by expressing microbially driven reaction rates per g cell biomass; then, the so-called specific rates (*q_s_* and *q_p_*) of cellular growth kinetics can be used as equivalents of the respective enzymatic rates, *v* [14]. For example, the Michaelis–Menten equation can be applied to the process of nutrient uptake by cells:(1)$$v=-\frac{ds}{dt}=V_{max}\frac{s}{K_{s}+s};\;V_{max}=k_{cat}\left[E\right]$$
where [*E*] is the concentration of the transmembrane transporter(s) responsible for uptake.

Dividing both parts of Equation (1) by cell mass *x*, we get the expression for SUR:(1a)$$q_{s}=-\frac{1}{x}\frac{ds}{dt}=Q_{s}\frac{s}{K_{s}+s};Q_{s}=\frac{V_{max}}{x}$$

The mass–balance relationship between the consumed substrate -*ds* and the produced biomass *dx* is set up by the yield, *Y*:(2)$$Y=-\frac{dx}{ds}=-\frac{1}{x}\frac{dx}{dt}:\frac{1}{x}\frac{ds}{dt}=\frac{\mu }{q_{s}}∴q_{s}=\frac{\mu }{Y}$$

The substitution of *q_s_* in Equation (1a) for *μ* produces the well-known Monod equation: (3)$$\mu =Yq_{s}=\mu_{m}\frac{s}{K_{s}+s};\;\mu_{m}=YQ_{s}$$

The Michaelis–Menten and Monod equations look similar, but there is a substantial difference. The enzymatic Equation (1) was deduced using the law of mass action and assuming the reversible formation of the enzyme–substrate complex as a unique mechanistic feature of enzymatic catalysis. It fits most experimental data, and the constants *V_max_*, *k_cat_*, and *K_m_* can be found for individual enzymes from UniProt, BRENDA, or other reference sources. What about the Monod equation? Unfortunately, the parameters *μ_m_*, *Y* and *K_s_* cannot be found in the NCBI taxonomy browser or any other reference source as trustworthy reproducible constants. There are many reasons for that but from a kinetic standpoint, the most essential flaw of the Monod equation is that it tacitly assumes the constancy of enzyme concentration, the variable [*E*] in Equation (1). This assumption is acceptable for a short-term response kinetic data obtained with either pure enzymes or intact cells but wrong as applied to microbial growth that is usually accompanied by significant MMCC changes. Figure 1 illustrates the phenomenon: the instant response of glucose-limited chemostat culture to a series of glucose pulses does fit Equation (1a) at each dilution rates *D (*Figure 1, Left). However, two parameters, *Q_s_* and *K_s_* were never remaining the same displaying a general trend to increase with *D* (Figure 1, Right). In Section B.3.2, we reproduced this pattern by simulating differential expression of low (L) and high (H) affinity glucose transporters. Every *D*-shift changes the ambient glucose concentration inducing the MMCC self-adjustment resulted in the optimized L:H ratio.

Other equations of enzyme kinetics (reviewed in [15]) are also expected to display a reasonable agreement with the short-term data on microbial populations but fail badly over longer periods, when microbial growth is accompanied with MMCC changes. Thus, it is the MMCC self-regulation that makes cell growth kinetics essentially more complex as compared with enzyme reactions.

### 2.3. Growth Conditions, the Hierarchy of Factors

The factors affecting growing cells can be physical (pressure, temperature, viscosity, stirring intensity, and mass transfer); chemical (impact of various chemical compounds); and biological (cell-to-cell interactions, differentiation into active and dormant, biofilm and planktonic forms, etc.). The biological factors are excluded from further discussion since they are parts of the dynamic phenotype to be the output rather than the input data in any growth model. The physical and chemical factors can be further split into (1) external independent variables, such as concentrations of nutrients or drugs, temperature, stirring rate, preset pH, etc., and (2) internal dependent variables, the environmental changes produced by growing cells, e.g., autoinhibitory products, metabolic acidification or alkalinization, production of ROS or RNS, biogenic heat, etc. An account of the first group of factors as the instant microbial response is straightforward [14,15,23,24,25]. The long-term response simulation requires an explicit account of the variable MMCC (see below ‘Synthetic chemostat model’ and Section B.3.3). Modeling of the growth side effects is more challenging but still a doable task, for example, metabolic acidification can be simulated by coupling the ODEs for *x* and *s* with the uptake and release of the major ionized compounds (NH_4_^+^, HCO_3_^−^, and organic acids) affecting SID, the strong ion difference [26].

Most factors play a modifying role, i.e., they can accelerate or slow growth but never can start it without one essential primary factor that is the nutrient supply. The nutritional factor is multidimensional; it includes a spectrum of consumable substrates; the identity and concentration of the limiting substrate; and the regime of supply: single-term or continuous; the continuous regime can be steady or periodic (diurnal, sinusoidal, square wave); or randomly fluctuating. Sometimes, all factors are viewed as equally important, depending on local circumstances, e.g., the temperature regime is supposed to be a key factor in cold environments like polar sea and the tundra, water deficiency in deserts, nutrient supply in oligotrophic lakes, etc. [27]. However, the nutritional factor stands alone as the only one that plays the role of a consumable resource rather than the state of the environment (T, pH, red-ox, conductivity, etc.) associated with the other factors. Another unique quality of the nutrients especially important for system biology and GEMs is that nutrients participate in growth stoichiometry and are subjected to the mass conservation law. First, the consumed nutrients are reactants in those primary metabolic reactions that initiate the entire metabolic network, eventually producing new cell masses and products. Any quantitative (concentrations) or qualitative changes in nutrients (medium composition) affect the status of M- and E-matrixes in GEMs. Second, a consumable resource is always limited (actually or potentially) to be shared between various competitors: coexisting species in communities; subpopulations in pure culture (e.g., planktonic vs. attached, mutant vs. WT, prototroph vs. auxotroph, etc.); or alternative metabolic pathways at network branching points. The most delicate goal of any GEM is to mimic the natural regulatory mechanisms to recognize the optimal alternative metabolic pathway. Thus, the nutrient factor is more elaborate and more challenging in mathematical simulations, eventually giving a precious reward of more insightful models improving the understanding of dynamic systems. Third, the temperature and other modifying environmental factors are important; however, their exploration can be meaningful only in combination with the nutritional factors. The reason is that microbial stress responses to nonoptimal physicochemical factors such as heat shock, freezing, acidity, etc. include the expression of chaperones, porins, and other molecular structures minimizing the cellular damage. As any other biosynthetic reactions, antistress metabolic responses require a supply of energy and anabolic substrates; therefore, they depend on nutrients. Interestingly, the maximum of the stress response is found not with a plentiful supply of nutrients but under substrate limitations [28]; the explanation is presented below (see the section on SCM (Section 4.2)). The opposite is not true; the effect of nutrients can be fruitfully explored separately from the other factors, providing they remain constant.

### 2.4. SGR and MMCC, the Chicken and the Egg Dilemma

A growing number of studies now deal with the MMCC changes associated with variations of SGR. The results are published under titles like ‘Effect of growth rate on expression...’ or ‘Growth rate dependent expression...’ [29,30,31,32,33,34,35]. Taking it literally, the time derivative of a cell mass concentration, *dx*/*dt*, plays a rather unusual role of a factor like pH, temperature, drug dose, and other independent variables typically characterizing external to cell environmental conditions. Let us see where the problem lies using as representative example the rRNA, one of the best-studied and the most abundant conditionally expressed macromolecule.

**The anatomy of experimental approaches.** There are two ways of experimental manipulation SGR: (i) a chemostat culture at a series of dilution rates attaining at a steady state the required condition, *D = μ*, and (ii) a batch cultivation using a series of C-sources supporting different SGR. The first technically more challenging approach has the advantage of keeping the same medium composition and a wider range of the tested SGR (typically, 0.05–0.95 *μ_m_*). The simpler batch method provides just several discrete SGR levels, e.g., in the range 0.4–1.8 h^−1^ for *E. coli* [36]. The essential difference is that microbial batch growth always remains substrate-sufficient (*s*
≫
*K_s_*), while the chemostat reproduces various degrees of substrate limitations. Therefore, cells grown batchwise on acetate (*μ* = 0.38 h^−1^) are not identical to cells maintained in a glucose-limited chemostat at *D* = 0.38 h^−1^. The physiological state of the nutrient limitations is preferential for diverse ecological and biomedical studies because in situ (soils and aquatic communities) and human body in vivo (gut microbiome and infection) microbial growths are typically strictly restricted by nutrients [22,28,37,38].

What are the independent and dependent variables in these two types of SGR manipulations? The independent variable is defined as a factor manipulated by the experimenter. The batch method implies manipulation with a medium composition, while the SGR computed from the growth curve is the dependent variable. With the chemostat method, the truly independent factor is also not SGR but the limiting nutrient concentration manipulated by the experimenter by setting up the dilution rate. The desired SGR is established spontaneously in a microbial culture over a transient period that takes hours and days until the steady-state condition *μ = D* is achieved. In other words, the experimenter does not have the power to directly manipulate the growth rate. The *D* management, such as medium flow increase, has an immediate effect on cells through the instant rise of *s*(*t*), the residual substrate concentration, the only environmental signal sensed by microorganisms that induces their self-adjustment through differential gene expression. Thus, the SGR is the dependent variable in both methods. Note that condition *μ = D* is not absolutely guaranteed even at a steady state because of the easily overlooked effects of wall growth and cell mortality (*μ < D*) or long-term spontaneous mutations with selection (*μ > D*) or sustained oscillation of variables *x* and *s* making SGR fluctuate around *D*. Sometimes, at one *D*, we observe two or more steady states as a memory effect of the prior growth conditions, a phenomenon called bistability [39].

**Causation test based on molecular data**. Now, we return to the ‘SGR-dependent rRNA content’. A typical experimental result is a linear plot of rRNA vs. SGR with a high correlation coefficient, but correlation does not imply causation. The options are: SGR → rRNA—the SGR is the causative factor for the variable rRNA content.rRNA → SGR—reverse causation.*s* → rRNA and *s* → SGR—both variables are affected by substrate concentration, *s*.
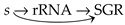
—options 2 and 3 are combined.

To select the best option, let us overview the published data on the rRNA expression [40,41,42]. In brief: each of the seven *rrn* operons in *E. coli* is transcribed by RNAP, the expression is inhibited by the alarmone ppGpp in response to uncharged tRNA (stringent control) and activated by NTP and indirectly by free amino acids (AA) that keep tRNAs charged. Although metabolomic data are limited and rather controversial (Section B.2.3 and Section B.3.3), there is evidence that intracellular pools of NTP and AA (two positive RNAP effectors) positively correlate with *s* (e.g., glucose), while the negative effector (p)ppGpp correlates with *s* negatively. It means that two options, 3 and 4, are justified, and *s* can be considered as the common independent factor affecting the SGR and rRNA expression (the SGR dependence on *s* was discovered 80 years ago [43] and is beyond any doubt). Finally, all the studied microorganisms, prokaryotic or eucaryotic, uniformly follow the so-called ‘ribosomal growth law’, which states that the ribosomal content in cells is the principal internal bottleneck [16,17,44,45]; the translation process is recognized as the slowest biosynthetic step; hence, growth acceleration can be achieved only through an increase of the ribosomal copy numbers. In other words, the SGR depends on the rRNA level, not the opposite! Thus, the rRNA expression data are in a full agreement with the previous section: the limiting substrate concentration in the chemostat is the principal independent factor; it affects the SGR directly (accelerating transport and rising the M-pool) and indirectly through the upregulation of rRNA. Options 2 and 3 are correct but incomplete, while option 1 is unconditionally wrong.

The conclusion derived using rRNA as an example is also applicable to other elements of conditionally expressed MMCC, even if they decline rather than increase with the SGR. The discussion is given below (Section 4.2 and Section 6.5 and Section B.3.2).

Why is incorrect causation SGR → rRNA so persistent? Probably, it became a part of the commonly accepted lexicon introduced in the middle of 20th century independently by several outstanding scientists [46,47,48]. The cliché ‘growth-dependent gene expression...’ conveys a clear message. Afterall, we economize on words speaking about speed-dependent fuel consumption by a car, although it is against the well-known causation sequence: fuel injection → combustion → … → car speed. Today, the words ‘growth-dependent’ are often replaced by ‘growth-associated’, a neutral wording avoiding any causation issue, and it is enough. The only exception presents mathematical models that must follow the causation matters as strictly as possible.

**The misrepresenting of SGR as an independent variable in mathematical simulations.** Various models, from very simple [32] to advanced GEM [49,50,51], use functional dependencies like *y* = *F*(*μ*), where *y* is a conditionally expressed variable like the rRNA content, and *F*(*μ*) is a simple (linear, hyperbolic, or exponential) function of SGR. In fact, *F*(*μ*) represents mathematically the already discredited option 1 above. What is wrong in using it in GEMs and other mathematical models? The SGR is attributed to the whole cell, integrating the contributions of numerous subcellular parts, including individual conditionally expressed RNA and proteins. The whole is a very awkward factor for predicting its parts because of circular references of the kind: SGR → y_1_ → … → y_n_ → SGR. Furthermore, SGR is the internal cellular characteristic that cannot convey any specific environmental signal affecting the phenotype. It resembles a hopeless Munchausen attempt to pull himself by the hair out of the swamp:



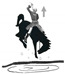



Newtonian mechanics do not allow such a process, because the rider and horse as a system at rest need an external force to move up the center of the mass. By the same token, a growth model using *F*(*μ*) functions ignoring the external growth conditions misses the processes’ driving forces; therefore, it is unable to reproduce microbial biodynamics as a self-regulated and self-evolving process.

Furthermore, implementing the function *y* = *F*(*μ*) is against the basic principles of the regulated gene expression, aimed to increase the versatility and adaptability of an organism to a changeable environment. It allows the cell to express only those gene products (proteins and RNA) that are required. For example, the lac operon turns on lactose metabolism only in the presence of lactose and absence of glucose. The function *y = F*(*μ*) does not carry any information about the state of the environment. SGR can be below *μ_m_* for many reasons (deficiency of nutrients, toxins, nonoptimal T, pH, etc.), and the model cannot predict based on the *y* = *F*(*μ*) function the way to optimize the expression profile.

### 2.5. Drawing the Line between Simple and Complex Biodynamics

At this point, we are ready to formulate a distinction between simple and complex microbial growths. A simple growth must be exponential with a constant SGR and a fixed MMCC that can occur only under steady environmental conditions (constant physicochemical and nutritional conditions and lack of self-poisoning). In a continuous culture, all the listed conditions are met when the system attains a steady state. Growth with constant MMCC has been already labeled by the term ‘balanced’ [52]. Thus, the epithets simple, balanced, and steady state can be applied as interchangeable synonyms. Three conditions (exponential pattern, constancy of SGR and MMCC, and steady environment) are coupled to each other, and if one of them is violated, then the other conditions also become invalidated. The linear growth observed in a fed-batch and perfusion culture seems to be simpler than the exponential, but it is not true, because it implies a progressive decrease of SGR that should be accompanied by respective MMCC changes; therefore, the linear growth is complex. Unsteady environmental conditions inevitably induce SGR and MMCC variations, and, finally, it is impossible to sustain a constant SGR under variable MMCC and vice versa. The complex biodynamics do not have any restrictions; they are observed under changeable conditions and accompanied by parallel changes of the growth rate and the MMCC status, including the expression profile. It is the most natural dynamic state of microorganisms in laboratory, industrial reactors and outdoor ecosystems. The next step of increased complexity is the spontaneous differentiation of cells into two or more subpopulations with the opposite traits, e.g., active–dormant, sessile–motile, attached–planktonic, etc. Below, we are going to demonstrate that a described dynamic complexity does not exclude the possibility of control and monitoring the state of the bioprocess as accurately and reproducibly as it is done with a simple growth.

## 3. Typical Examples of Simple and Complex Biodynamics

### 3.1. Batch Culture

The canonical pattern of the batch growth curve as a sequence of the four phases (Figure 2A) becomes an indispensable illustration reproduced in every microbiology textbook. We traced the origin of this sketch back to 1918 [53]. Robert Earle Buchanan (1883–1973) was one of the most influential bacteriologists worldwide, contributing most significantly to the bacterial taxonomy. In 1918, he was 35 and already eight years in charge of bacteriology and the newly appointed Dean at Iowa State College, “…hard to think of him working at a bench” [54]. Accordingly, his lucky paper was not just a routine experimental report; rather, it was a generalized summary of the accumulated knowledge on bacterial development analogous to the ontogeny of plants and animals. The paper ignores methodological details and presents growth curves as trends without actual data points. To feel the gap, we addressed the contemporary relevant publications [55,56,57,58] and found that bacteria were grown that time on broth (mostly brain infusion) in tubes without stirring or pH control, and cell concentration changes over time were assessed by plating. The real published data differed from the Buchanan profile by two main points: (i) the absence of a stationary phase as an extended period when ‘the growth rate is equal to the death rate’ [53]; instead, all growth curves passed the sharp peaks with the following decline and (ii) a more extended death phase as compared with the Buchanan pattern. Growth on proteinous broth is accompanied by the release of NH_4_^+^ and severe alkaline stress (pH curve added to Figure 2A based on our studies), which is another reason for growth halt in addition to the nutrient’s depletion. Batch growth under defined conditions (Figure 2B) further confirms two mentioned inadequacies. The exponential phase is immediately followed by the starvation death phase with a progressive decline of the mortality rate over time indicating a self-regulatory stress-response of the starving bacteria.

Batch growth limited by an anabolic (conserved) substrate (Figure 2C) deviates most significantly from the canonical version; it does not stop after depletion of the limiting substrate (phosphate), and cells continue division without nutrient consumption. The entire growth curve is unbalanced and complex. Finally, the fragment of the batch process associated with the biosynthesis of secondary metabolites, such as antibiotics (Figure 2D), is completely beyond the exponential phase of the balanced growth, since the biosynthesis of secondary metabolites is always activated in nearly depleted cultures.

To summarize, the simple balanced growth in a batch culture is either absent or can be found only as a relatively short fragment of the entire growth dynamics in the culture limited by C- and energy sources. The stationary phase erroneously shown in every textbook is not confirmed using controlled cultivation; there could not be a balance between the growth and the death rates, i.e., a batch culture can never attain a steady state. However, at the exponential growth phase, the physicochemical conditions remain relatively stable, and the nutrients are not yet depleted, while the toxic products are not yet formed. This relatively short fragment of the entire growth curve is characterized by constant SGR and MMCC and can be qualified as a simple and balanced growth. Formally, the system does not comply with the steady-state conditions, since variables *x* and *s* are not constant, but another condition is fulfilled, the constancy of specific rates (1/*x*)(*dx*/*dt*) and −(1/*x*)(*ds*/*dt*). The instrumental pH control provides a longer simple growth with constant SGR and MMCC. The most intriguing and extended dynamic phenomena over time: unusual growth patterns on anabolic substrates, lag phase of a metabolic reconfiguration, starvation survival accompanied by cellular differentiation, and the formation of secondary metabolites all take place beyond the phase of simple balanced biodynamics.

### 3.2. Continuous Culture

The transient shift-up process after a stepwise *D* increase (Figure 3A) demonstrates the biological inertia of cells manifested as conspicuous overshoots and undershoots before attaining a new steady state. With successive small-step *D*-changes (Figure 3B), the transients are much smoother. The third type of hidden transients (Figure 3C) is associated with spontaneous beneficial mutation and selection sweeps, the displacement of a parental line with a mutant acquired a higher affinity to the limiting substrate (lower *K_s_*). The sweep is manifested as a noticeable decrease of *s*, as well as the interruption in the linear rise of the neutral mutations in phage resistance [61] or drug resistance (Figure 3C, open circles). The interrupted mutant dynamics mark a ‘clock resetting’, all cells are competitively eliminated, including the WT and resistant variants.

As compared with the batch culture, the chemostat allows improved control over microbial growth with an extended duration of the simple balanced and truly steady-state growth under a stable environment. Although we need to watch for a possible selection sweep and biofilm formation (wall growth), nevertheless, the culture stabilized at constant *D* is by far the best representation of simple growth. The transitions between steady states belong to the category of complex biodynamics. The non-steady-state chemostat has another remarkable advantage by fixing over time the starting and terminating states of the transient process, making non-steady-state growth highly reproducible and amenable to vigorous kinetic analyses. Importantly, the pooled chemostat data from several *D* no longer belong to the category of simple growth, even if all the transient data are excluded.

Indeed, at each *D*, the cells acquire unique MMCC and growth characteristics (substrate affinity, metabolic potential, yield, etc.) that are distinct from other *D.* These data can be combined under the umbrella of the complex growth model. Figure 3D illustrates the amplitude of growth-associated changes in the chemical composition of the cells; the proteome expression will be discussed below.

Figure 4 illustrates biodynamics in the continuous culture retaining growing cells (syn: perfusion culture, retentostat, and chemostat with cell recycle) that uses continuous dialysis or tangential filtration to return growing cells back to the cultivation vessel. Contrary to the regular chemostat, the retentostat attains a quasi-steady state: the cell mass *x* increases (as there is no washout), while *ds/dt* ~ 0. Over time, the nutrient supply per unit cell mass dramatically decreases, driving the population to a state of chronic starvation, an extremely deep substrate limitation but without the decay and self-poisoning typical for a depleted batch culture. That is why perfusion is especially beneficial for the cultivation of slowly growing mammalian cells and fastidious microorganisms. From the 10th day onwards, the division frequency drops down to as low as one division per month, comparable with the reproductive pace of natural microbial populations in the most challenging oligotrophic habitats (lakes, sea pelagic zone, soils, and subsoils) that are otherwise impossible to reproduce in the laboratory [28]. The near-zero growth provokes the formation of nonreplicative VBNC cells and deviates considerably not only from the simple growth discussed above but also from a more familiar complex biodynamic observed in a regular chemostat.

## 4. Pre-Genomic Models of Microbial Growth

### 4.1. From Malthusian Exponent to the Cybernetic Models

Now, the task is to overview the ability of mathematical models to adequately reproduce simple and complex microbial dynamics. We start from the pre-genomic models as predecessors of much more comprehensive genome-scale models. Table 1 presents the selected growth models, the milestones of microbial growth kinetics. The exponential model (4) has a historical significance as the first demographic model, albeit under a naïve assumption of unlimited growth. The next three models, (5)–(7), are more realistic, although still ignore the MMCC changes; they introduce various ways to restrict unlimited growth: the logistic Equation (5) assumes that growth is negatively affected by density-dependent biotic interactions, the Monod model (6) explains the growth restriction by the limited availability of nutrients, and finally, the Monod–Ierusalimsky model (7) considers the combined restrictive effect of substrate limitation and product inhibition. The Monod-type models (6) and (7) were already discussed above; they represent cells as ‘proliferating enzymes’ ignoring the possibility of self-regulatory changes of the biokinetic qualities of cells. Numerous attempts to improve the Monod equation, such as adding a third parameter or a second variable (e.g., *s* and *x*) and using an alternative to hyperbola expressions, were not successful; see the details in [22]. 

The rest of the models (8)–(11), allow MMCC variations, although to different degrees. The cell quota model (8) was developed for phototrophic microorganisms using light as an unlimited energy source, their growth being controlled by the availability of anabolic substrates, mostly sources of N, P, and vitamins. The cell quota, σ, is the variable content of the deficient element in a cell mass, and the respective growth model was the first attempt to explicitly express the relationship between the SGR and the chemical composition of the cells. The compartmental and structured models (9) are the closest relative to the GEMs, exploiting the idea to combine exchange fluxes (uptakes of exogenous substrates) with internal cellular variables. The simplest models contained just two to three highly aggregated internal variables (e.g., RNA, DNA, and proteins); the more advanced models covered up to 50 real metabolites and the lumped variables such as pools of amino acids, rNTP, dNTP, cell wall precursors, RNAs, DNA, ppGpp, glycogen, and peptidoglycan [63]. They never gained a wide recognition, remaining a caricature of the real metabolic network for biochemists, cumbersome ODE sets for mathematicians, and an awkward research tool for systems biologists. The obvious disadvantages were (i) inevitable subjectivity in the selection of the most important internal variables, and (ii) an unresolved problem of experimental verification of the incomplete models against the real biochemical data. The cybernetic models (10) were initially introduced as simple structured models, the word ‘cybernetic’ implying that living cells and electronic gadgets share the common feature of self-regulation. They used special cybernetic variables **v** and **u** as empirical Boolean functions to implement the principle of optimality like FBA models do now. More recent cybernetic models still use the **v** and **u** variables but are scaled up to the genomic level [64,65,66,67]. Their value as a special category of GEM is questionable because of the artificial nature of the operational cybernetic variables. The negative feature shared by all three (Equations (8)–(10)) models was that they mostly ignored the growth conditions and did not systematically follow the relationship between the growth and MMCC. The only exception was the Synthetic Chemostat Model (SCM; Equation (11)) that is described below in more detail.

**Table 1 microorganisms-09-02352-t001:** Pre-genomic models of microbial growth kinetics.

ODE	Equation	Comments
$\frac{dx}{dt}=\mu x$	(4)	Exponential growth [68]. SGR is constant
$\frac{dx}{dt}=Rx\left(1-\frac{x}{R}\right),\;R=\mu -a$	(5)	Logistic equation [69]. Growth is restricted by the negative biotic interactions; *K* is the upper *x* limit
$\begin{array}{c} \frac{dx}{dt}=\mu x-Dx;\frac{ds}{dt}=D\left(s_{r}-s\right)-\frac{\mu x}{Y}-mx;\\ \mu =\mu_{m}\frac{\left(s-s^{*}\right)}{K_{s}+s},\;s^{*}=\frac{K_{s}mY^{max}}{\mu_{m}},\;Y=-\frac{dx}{ds}=\frac{Y^{max}D}{D\;+\;mY^{max}} \end{array}$	(6)	Monod chemostat model [70] with a maintenance term [14]. SGR is a hyperbolic function of *s* and is negative below threshold *s*.* The yield *Y* varies because of the maintenance
$\frac{dp}{dt}=Y_{p}\mu x-Dp;\mu =\mu_{m}\frac{s}{K_{s}+s}\frac{K_{p}}{K_{p}+p}$	(7)	Monod-Ierusalimsky model [22] accounting self-inhibition by the product, *p*; *x* and *s* are defined by (6)
$\begin{array}{c} \frac{dx}{dt}=\mu x-Dx;\frac{ds}{dt}=D\left(s_{r}-s\right)-\mu x\sigma ;\\ \mu =\mu_{m}\;\frac{\sigma_{m}\left(\sigma -\sigma_{0}\right)}{\sigma \left(\sigma_{m}-\sigma_{0}\right)} \end{array}$	(8)	Droop model [22,71]. The ‘cell quota’ σ is the content of deficient nutrient element in cells. The model was specially designed for microbial growth limited by the conserved nutrient substrates
$\begin{array}{c} \frac{dC_{i}}{dt}=r_{1}\left(s,C_{1},\;\ldots ,\;C_{n}\right)-r_{2}\left(s,C_{1},\;\ldots ,\;C_{n}\right)-\mu C_{i}\\ \sum_{i=1}^{n}C_{i}=1 \end{array}$	(9)	Structured models with multiple internal variables *C_i_* [72]. Model includes two types of processes: (i) exchange reactions between cells and surrounding milieu (nutrients uptake, products release, see Equations (3)–(5)) and (ii) the MMCC changes
$\begin{array}{c} \frac{dx}{dt}=x\sum_{i=1}^{2}\mu_{i}v_{i}-Dx;v_{i}=\frac{\mu_{i}}{\max\left(\mu_{i}\right)}\\ \frac{ds_{i}}{dt}=D\left(s_{ri}-s_{i}\right)-x\left(\frac{\mu_{i}v_{i}}{Y_{i}}+m_{i}v_{mi}\right)\\ \frac{de_{i}}{dt}=q_{i}u_{i}-\mu e_{i};u_{i}=\frac{\mu_{i}}{\mu },\;\mu =\sum_{i=1}^{2}\mu_{mi}e_{i}\frac{s_{i}}{K_{si}+s_{i}} \end{array}$	(10)	Cybernetic models [73] describing the diauxic growth of bacteria on the mixture of glucose and lactose taken up by transporters e_1_ and e_2_ respectively. The cybernetic variables regulate transcription (*u*) and uptake rates (*v*), e.g., $v_{1}=\;\left\{\frac{1,\;\;\mathrm{if}\;\;\mu_{2}\;<\;\mu_{1}}{0,\;\;\mathrm{if}\;\;\mu_{2}\;\geq \;\mu_{1}}\right.;\;v_{2}=\;\left\{\frac{0,\;\;\mathrm{if}\;\;\mu_{2}\;<\;\mu_{1}}{1,\;\;\mathrm{if}\;\;\mu_{2}\;\geq \;\mu_{1}}\right.$
$\begin{array}{c} \frac{ds}{dt}=D\left(s_{r}-s\right)-q_{s}x,\;q_{s}=rQ\frac{s}{K_{s}+s}+\left(1-r\right)\frac{Q^{\prime }s}{K_{s}^{\prime }+s}\\ \frac{dx}{dt}=\mu x-Dx,\;\mu =Yq_{s}-a_{0}r\;\\ \frac{dr}{dt}=\mu \left(\frac{s}{K_{r}+s}-r\right)\; \end{array}$	(11)	Synthetic Chemostat Model [74]. The *r*-variable integrates the MMCC variation and tunes the kinetic terms earlier assumed to be constants (*m*, *Q*, *a*, and *K_s_*). The ODE *d**r*/*dt* explains the observed biological inertia. See the Section 4.3 for other details

See the list of common symbols in Appendix B. Red marks the regulatory self-adjustable variables in the cybernetic models and SCM.

### 4.2. Synthetic Chemostat Model (SCM)

The SCM has been built upon all the chemostat models listed in Table 1 (that explains the epithet synthetic). It pragmatically denies the likelihood of the explicit accounting of thousands of individual MMCC and instead splits them into the P and U clusters with opposite expression trends (Table 2).

Each component has low (*P_i_*^min^ and *U_j_*^min^) and high limits (*P_i_*^max^ and *U_j_*^max^). Normalized to the total biomass *x* vectors, P and U are bound by the conservation condition P + U = 1 (for simplicity, a minor contribution of the small molecules is neglected). Then, any increase in the sum of the P-components with growth acceleration should be compensated by an equivalent decrease of the U-sum and vice versa. The heart of the SCM is the assumption that the sum of the two vectors P + U can be represented as the following linear function of one scalar variable *r* and three constant vectors:



(12)

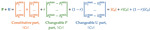




The vectors |*C*_0_|, |*C_P_*|, and |*C_U_*| remain constant for a given organism under any growth conditions. The *r*-variable depends on *s* Equation (11), and other factors (T, pH, and osmolarity) can be added in a multiplicative way. The role of the *r*-variable is to be a reporter on the status of the entire MMCC. The higher the *r*, the higher the expression degree of the P-components supporting intensive growth, with a penalty for reduced stress resistance. The *r*-variable participates in the rate expressions for SUR, SGR, turnover, and the maintenance term, providing a higher flexibility in response to the growth conditions. More technical and interpretive details about the SCM are given in Section B.3.3.

### 4.3. Strength and Limitations of SCM

The main strength comes from introducing the link between the growth kinetics and the state of the MMCC. It explains a long list of the formerly mysterious behaviors repeatedly observed in the cultivation practice (Table 3). Despite its simplicity (just three ODEs in the basic model and a few more in specialized versions), the SCM adequately simulates a wide range of complex microbial behaviors. All the dynamic curves plotted in Figure 1, Figure 2, Figure 3, Figure 4 and Figure 5 were calculated by using a SCM demonstrating an adequate agreement with the observations. The principal model limitation stems also from its simplicity, giving the disadvantage of low resolution in the interpretation of omics array-style molecular data. This deficiency can be fixed by integrating the SCM with the appropriate GEM.

How reasonable is the assumption expressed by Equation (12)? Essentially, it implies that diverse individual macromolecules vary in response to environmental stimuli in a coordinated synchronous fashion, e.g., if one P-component starts upregulation, then the other should follow the same trend, while U-components should be downregulated to comply with the conservation condition. What is the biological logic behind this synchrony? We list below four direct and indirect supporting arguments.

First, a commonly accepted empirical fact discovered about 100 years ago [75] states that stress resistance is minimal under optimal growth conditions. This empirical generalization has never been rejected. Second, the synchrony between various P-components is expected based on molecular stoichiometry. For example, the ribosomes in *E. coli* contain 55 r-proteins and three rRNAs, which are associated with elongation factor Tu, 21 tRNAs, and the tRNA synthetases charging tRNAs)—all these ~100 giant macromolecules are tightly synchronized by several mechanisms to sustain the uninterrupted synthesis of proteins [41]. Third, transcriptional control of the expression in bacteria is regulated by sigma factors for an extended set of genes rather than for a single gene. The number of sigma factors and their functions vary among taxonomic groups of bacteria; in *E. coli,* the factors σ^S^ activated under one particular stress, say, starvation, initiate a synchronous expression of up to 100 genes responsible for resistance to various stresses [76]. Similar patterns of stress response were observed in eukaryotic cells, in *S. cerevisiae*, a stress response affects the expression of ~900 genes [77]. Fourth, the most convincing and direct evidence comes from the proteomic data obtained for *E. coli* grown in a glucose-limited chemostat [21]. After normalizing and replotted vs. SGR, the original data formed two clusters of conditionally expressed proteins behaving in remarkable similarity to the main SCM postulate (Equation (12), Figure 5). Thus, the proteome of *E. coli* is conditionally expressed in coordinated synchrony, indicating that SCM correctly conveys the general trend in gene expression.

## 5. Genome-Scale Models

The first organism to have its entire genome sequenced was *Haemophilus influenzae* in 1995 [78], followed by *E. coli* K-12 and several other species in 1996 to 1997 [79]. The respective genome-scale metabolic reconstructions of these bacteria were completed in few years [80,81]. From that time on, the number of GEM publications has risen exponentially [1,82,83]. This progress would not be possible without the pre-genomic period of 1990–1999 when the constraint-based modeling approach was developed and tested using incomplete genomic data [84,85,86,87,88,89,90,91]. The San Diego team led by Bernhard Palsson made overwhelming contributions to the process with prolific publications, conference talks, developed free public software accompanied with user-friendly tutorials [36], and easy-to-learn textbooks [12,92,93]; not to be forgotten, numerous alumni have graduated from his SDSU Center and went into academia and the bioindustry across the world to form new hotspots of GEM-based research.

### 5.1. GEMs Cover Only the Part of the Genome

Some published GEMs represent as low as 2% of the bacterial genomes simulating the core/central metabolism [94]. The most complete metabolic, GEM [95], covers up to 35% of the genome, and it is almost the top. Figure 6 explains why the coverage cannot be higher. From a modeling perspective [92], there are four distinct parts of the genome: (i) junk material and sequencing errors need to be removed (blue background); (ii) the M-matrix (M for metabolism) stands for genes encoding enzymes and catalyzing the entire intermediary metabolism, which is about one-third in prokaryotes and a smaller fraction in eukaryotes [10]; (iii) the E-matrix (E for expression) accommodating genes encoding the enzymes and RNAs responsible for proteins synthesis, transcription, and translation; and (iv) finally, the O-matrix (O for operon) containing the transcriptional regulatory network, a part of the global regulatory machinery orchestrating cellular functions via TFs. The respective GEMs are called M, E, and O models. The reconstruction of metabolic networks (M models) is now a well-established process (see the below FBA and dFBA models). The E models have recently been started by using a similar approach and are produced as combined ME-models. Both models convert genomic data into stoichiometric matrices of intermediates (M-matrix) and biosynthetic reactions (E-matrix) that are made interactive. Both models apply common optimality assumptions and constraints, as well as a quasi-steady state approximation, allowing their easier computational solution. Unfortunately, there is no way to predict in theory or based on sequence similarity the functionality of the O-matrix, ‘the brain’ of the metabolic network. Therefore, merging the O module into ME models has been claimed [10] to be impossible. However, a fully integrated OEM model was built for the simplest parasitic prokaryote [96], called a whole-cell (WC) model. Recently, WC models were developed for *E. coli* and *S. cerevisiae*, although without addressing the whole genome.

### 5.2. Flux Balance Analysis (FBA)

This technique has been introduced as a special case of the simplified genome-scale metabolic reconstruction [12,36,92,93,97,98,99,100]: (i) it seeks only steady-state solutions for metabolic fluxes; otherwise, it would be necessary to define the reaction kinetics and the metabolic pool size for hundreds of intermediates, which is unrealistic; (ii) the constraints are used to define a closed solution space for the flux vectors; commonly used constraints include the feasible ranges of each metabolic flux, and additional constraints come from reactions thermodynamics, as well as from the metabolomic, transcriptomic, or proteomic data; (iii) finally, the best solution is found using linear optimization for a certain objective function.

There are both advantages and limitations of the FBA. The main advantage is its computational efficiency: the COBRA toolbox in MATLAB [101] or COBRApy in Python [102] installed on a regular laptop executes full-size M models in seconds. Other advantages include (i) the uniform structure of FBA that is potentially applicable to any sequenced organism after the manual curation of genomic data, including the non-model newly isolated bacteria [103], (ii) the unbiased genome-scale presentation of cellular metabolism making redundant the selection of the ’most important reactions’ as in the pre-genomic structured models, and (iii) a good agreement with the available experimental data [36]. The FBA is not designed for dynamic simulations and cannot present growth as a time series of *x*, *s*, and *p* to demonstrate the agreement. Instead, it produces the metabolic map with fluxes for each reaction. It correctly reproduces reconfiguration of the M-matrix after switching from one C-source to another, from aerobic to anoxic growth conditions; it also accurately predicts gene essentiality and the impact of gene knockdown on metabolic flows, including the exometabolic products especially valuable for bioengineering.

The limitations of FBA as admitted by the developers [36] include neglecting the kinetics of the metabolic reactions and their self-regulation. In Section B.1 and Section B.2.1, we provided an explicit solution of the FBA equations that show what specific kinetic data are required for calculation of the steady-state concentrations of metabolic intermediates. By using the introduced above growth terms, the FBA can deal with only the simple, balanced, and steady-state growth under constant optimal environmental conditions. The fixed MMCC assumes a simplified way to present the biomass as the product of a pseudo-reaction with constant stoichiometric coefficients *k*_1_*…k_n_*:(13)$$\begin{array}{ccc} \mathrm{intermediates}\;\mathrm{as}\;\mathrm{reactants} & & \mathrm{cell}\;\mathrm{biomass}\\ k_{1}c_{1}+k_{2}c_{2}+\ldots k_{n}c_{n} & ⟶ & C_{38.3}H_{67.9}O_{18.8}N_{9.2}Z_{1.1},\;n\sim 100\; \end{array}$$

Here, the ‘mole’ of biomass is based on the average elemental composition of cells taken from the mid-exponential growth phase. For *E. coli,* the left side of the reaction of Equation (13) contains as reactants 20 amino acids, 4 rNTP, 4 dNTP, 15 inorganic ions, 14 cofactors, and 3 precursors of peptidoglycan and lipids. Using the corrected biomass formulae for two phases of the diauxic growth of *E. coli* instead of a fixed one [88,89] substantially improved the FBA simulation; however, it was an ad hoc solution appropriate only for this model organism.

The FBA has claimed to predict the SGR based on genomic data. However, it is not perfectly true, because we must select the low boundary for the substrate uptake −*q_s_*. If it matches exactly the maximum uptake rate, *Q_s_*, then the generated SGR corresponds to the *μ_m_*. The *Q_s_* is to be found in publications or we have to run a simple batch experiment when recording the C-source substrate *s*(*t*) and the cell mass *x*(*t*) time series. Then, the nonlinear (exponential) regression produces three growth constants: *μ_m_*, *Q_s_*, and *Y*. In other words, if we enter the experimentally verified −*q_s_* or −*Q_s_*, we already know SGR without any genomic simulation! Note also that *μ_m_* is an easier parameter to measure than *Q_s_*. No doubt, the FBA capacity to reproduce *μ_m_* based on the specified *Q_s_* remains to be a nontrivial task, as it involves the computing of the huge M-matrix. If we accidentally or intentionally select *q_s_* < *Q_s_*, then FBA generates SGR proportionally decreased vs. *μ_m_*, but metabolic flux maps preserve the same pattern as in the case of unlimited growth. It means that the FBA does not reproduce the substrate-limited growth, as claimed by Reference [36], giving instead just a wrong measure of *μ_m_*.

### 5.3. The Dynamic FBA (dFBA)

These models apply a quasi-steady-state approximation. Microbial growth can be always viewed as a sum of processes with different characteristic times from extremely slow (microevolution in a chemostat culture, including selection sweep) to slow (cell biomass) and fast (metabolic pool, respiration, SUR, and limiting substrate concentration). The metabolic fluxes can also be differentiated into slow and fast categories, as dependent on the reaction volume [104]. The exchange reactions between the cell and environment (nutrients uptake and products release) take place in the space from ml (culture tube) to 10-L benchtop bioreactor, the volume range 10^0^–10^4^ cm^3^. The intracellular metabolic reactions occur in a much smaller volume of a single cell, 10^−8^–10^−9^ cm^3^. A volumetric difference by 10^8^–10^13^ times implies that intracellular reactions can be safely assumed to be at a steady state, and therefore, the flux distribution can be resolved by the FBA. Few remaining slow exchange variables are to be numerically solved as a set of ODEs. 

For the practical implementation of the dFBA [87,105], two approaches were developed: (i) the Dynamic Optimization Approach (DOA) by using NLP and (ii) the Static Optimization Approach (SOA) that discretizes the bioprocess time into small time intervals and uses LP at the beginning of each time interval. The agreement of dynamic simulation with experimental data should be estimated as modest (Figure 7). There were other numerous applications of dFBA to simulate the biodynamics of industrially important organisms: *S. cerevisiae* and *E. coli*, separately or as mixed culture [106,107,108,109], *Shewanella oneidensis* [110], *Chlorella* [111], CHO cells [112], *Pichia pastoris* [113], *Aspergillus niger* [114], and *Streptomyces tsukubaensis*, producing an immunosuppressive drug [115]. With rare exceptions, the mismatch between observation and prediction was large. Mostly, the failures occurred in attempts to go beyond a simple growth. It was expected since the exchange reactions were represented by the outdated Monod model.

### 5.4. The Modification of dFBA:rFBA and iFBA

The regulatory FBA models [94] dealt with the simulation of the central part of *E. coli* metabolism accounting for 149 genes, encoding 16 regulatory proteins and 73 enzymes, and 113 reactions, 45 of which were under transcriptional control, using Boolean functions. The model can predict the growth curves of *E. coli* on defined media, as well as the substrate uptake, metabolic byproduct secretion, and gene expression under variable growth conditions. The rFBAs were also combined with classic kinetic models based on ODEs [116,117]; the hybrid was called the integrated FBA (iFBA), and it gave a more accurate prediction of phenotypes. However, iFBA lacked generality and could be applied only to the well-studied model organisms. None of these FBA modifications has been applied to the complex biodynamics associated with MMCC variations. It was impossible by default, because the exchange reactions were presented by the Monod model.

### 5.5. ME-Models

The first ME model coupling metabolic processes with gene expression were developed for *E. coli* with the account of 423 genes [118]; it produced the long-waited result of the successful simulation of growth-associated MMCC self-regulation, demonstrating excellent agreement between the observed and simulated numbers of ribosomes. Thereafter, ME models were applied to the small-genome hyperthermophilic bacteria *Thermotoga maritima,* accounting for 651 genes [119], and to *E. coli* K-12 describing the synthesis and functions of almost 2000 gene products [49,120]. Then, the model was further developed to account for the spatial proteins allocation between the cytosol, periplasm, and inner and outer membranes [121]. 

The reconstruction was commonly recognized as being both difficult to compute and challenging to understand conceptually. Two improvements were soon offered: (i) the software product solveME provided up to 45% acceleration using a quad-precision NLP solver [122] and (ii) COBRAme [123], which condensed the original ME model around five times without a loss of resolution and functionality, reducing the computing time from 6 h to less than 10 min! COBRAme has been unified and reformulated into a software framework allowing to build and edit the ME models of any sequenced and annotated organism. Soon, it was applied to other organisms, including anaerobic mixotroph *Clostridium ljungdahlii,* a promising biofuel producer [124]. 

A recent review [1] followed step-by-step the progress in ME model development, including their growing coverage of genome, key findings, and applications. Below, we list their distinctive features as compared with FBA and dFBA based on three reconstructions designed for *E. coli* [49,123,125]. The model covered 1541 unique ORFs and 109 RNA genes, accounting for ~90% of the *E. coli*’s proteome. As compared with prior M models, the ME models face two major challenges. First, the dimension of the E-matrix is about 30 times bigger, and its elements differ by 15 orders of magnitude, presenting a severe ill-conditioned matrix problem. Second, contrary to the M models focusing on fluxome and ignoring reaction kinetics and metabolic pool sizes, the E-models are meaningless without the amounts (contents and concentrations) of conditionally expressed proteins; therefore, the kinetic characterizations and attempts to quantify concentrations are unavoidable. In practice, the ME-models were implemented as follows:


The metabolic network was treated similarly to FBA using constraints and objective functions. The maximizing growth rate was combined with minimizing biosynthetic costs of macromolecules. Substrate concentration was never shown as an independent variable; nutrient deficiency was accounted for by using the substrate availability bounds.The mostly unknown *k_cat_* values for translational and transcriptional enzymes were set up as a gross average 65 s^−1^. Considerable improvement in the simulation was achieved by combining the ME model with proteomic data as the model’s input [125]. It allowed an estimation of the individual turnover rates *k_eff_* for *E. coli*. The range of *k_eff_* varied within eight orders of magnitude, and surprisingly, it was not significantly impacted by variations of the cultivation conditions (four different C-sources).It can be shown (Section B.2.5) that the rate constant *k_eff_* is dependent on the concentration of the respective intracellular substrate, *M*, e.g., in the form of the Michaelis–Menten equation: *k_eff_* = *k_cat_M/*(*K_m_* + *M*). For unlimited growth, *M*
≫
*K_m_*, and *k_eff_* = *k_cat_*, while at the nutrient limitation, *k_eff_* < *k_cat_*. However, in vitro kinetic data for the isolated *E. coli* enzymes remain incomplete and most probably not adequately characterize the in vivo metabolic processes (Section B.3.5). Instead of using published data, the authors applied a convoluted two-steps binary search procedure involving an auxiliary ‘dummy protein’ computing (first step) followed by the second binary search to find the minimal ratio *k_eff_/k_cat_* as a condition making dummy protein formation to be zero.The rates of transcription and translation have been assumed to follow the empirical hyperbolic function of SGR, e.g., the translation rate $v=V^{m}\frac{\mu }{K+µ}$.The biomass reaction, Equation (13), was completely removed, because most of the components (amino acids, nucleotides, etc.) were already accounted for in the proteins and RNA biosynthesis reactions. Instead, the reduced pseudo-reaction was used for the remaining cell constituents: glycogen; DNA; lipids; peptidoglycans; and several cofactors (NAD, coenzyme A, etc.). The expressed but not incorporated into the model proteins were called the unmodeled protein biomass. Most likely, these proteins are responsible for stress resistance (U cluster in SCM). This fraction has been also added to the modified biomass reaction to comply with the growth–mass balance but excluded from discussion of their biological role.The only sink for macromolecules was assumed to be their dilution caused by growth, so any metabolic rate *v_i_* and translation rates of the respective enzymes were set up as follows:




(14)
$$v_{translation,i}=v_{dilution,i}=\mu \left[E_{i}\right];\;v_{i}=k_{eff,i}\left[E_{i}\right]\;∴\;v_{translation,i}=\frac{\mu }{k_{eff,i}}v_{i}$$



The constraining condition providing the *v_i_* flux sustainability was [125]:(15)$$v_{i}\;\leq \;\frac{k_{eff,i}}{\mu }v_{translation,i}$$

Some of the listed assumptions are not justified, specifically, the points B and C (replacing Michaelis–Menten or Hill enzyme kinetics with constant rates found via a binary search) and D (using SGR as an independent variable, discussed above in Section 2.4). Appendix B (Section B.1 and Section B.2) summarizes the published *E. coli* metabolomic data. We conclude that 90–95% of metabolic reactions within the M-matrix are not saturated by the available intracellular substrates (*M*
≪
*K_m_*), validating the first-order rather than zero-order approximation. The remaining 5–10% of the metabolic reactions follow the mixed kinetic order, and only a few out of the +5000 reactions approach the state of substrate saturation (zero order assumed by the point B). The inconsistent assumptions made in points B–D could be the reason for the failure of ME models to correctly reproduce a proteomic profile of *E. coli* in the chemostat culture (Figure 5C). 

How successful are ME simulations? Unfortunately, none of the published ME models demonstrated an adequate dynamic simulation of a microbial growth and expression profile (transcriptomic or proteomic) under any cultivation scenarios shown above by Figure 1, Figure 2, Figure 3, Figure 4 and Figure 5. On the other side, M models were able to simulate the perturbation experiment in the batch culture of *E. coli* [125]: bacteria were grown on the minimal glucose–mineral medium and then the second C-source was added (adenine, glycine, tryptophan, or threonine). The expression patterns were 56–100% correctly predicted, but it was a single point of the semi-quantitative simulation, leaving unanswered questions about the dynamic growth pattern. Another successful application was the model OxidizeME [126] simulating an *E. coli* response to oxidative stress, including: (i) ROS-induced auxotrophy for several unstable amino acids, (ii) a nutrient-dependent variation of stress resistance, and (iii) ROS-induced differential gene expression. The only regret is that the OxidizeME has been customized to accurately reproduce as closely as possible the already known mechanisms and only those affecting iron–sulfur clusters. There are numerous other targets for ROS, and there are many other stresses. We want GEM to assist us in finding the common and specific effects of diverse stress factors, as well as to predict (based on the genome) the dynamic pattern of the microbial stress response, including the progress of recovery.

What is the impact of ME models on modern system biology? Every publication brought some novel results of general biological significance: the role of multiple *rrn* operons [118], synonymous codon usage bias [120], revised interpretation of the growth limitations [49], translational pausing regularities, and synchronization of the cellular synthesis of macromolecules [125].

## 6. Whole-Cell Genome-Scale Simulations (WC Models)

This type of GEM should be a *summum bonum* of microbial system biology. Ideally, all parts (O, E, and M matrices) of the genome must be accounted for in the simulation. However, it is not realistic now, because the functions of many genes are not yet defined, and we know too little about the global regulation of cellular machinery, how is it orchestrated via the O-matrix or otherwise. Three partially resolved WC models are discussed below.

### 6.1. Mycoplasma genitalium

The first modeling trial was applied to the free-living parasite bacterium *Mycoplasma genitalium* with the smallest genome accommodating 525 genes, including 382 essential genes [96,127,128]. Even the simplest prokaryotic cell was an exceptionally challenging object for simulation; the problem was solved by using the modularity approach [129]: cellular machinery was split into 28 modules (transcription, translation, metabolism, replication, DNA repair, host interactions, etc.), each of which was solved independently at the integration step 1 s. The modules were represented by ODE and other equations with discrete and continuous variables, mechanistic and empirical (not fully specified in the publication). The most advanced computer technique was implemented (a 128-core Linux cluster), and the simulation took 10 h for a single *M. genitalium* cell to divide once, about the same time the actual cell takes. What have we learned from these remarkable modeling efforts? The authors listed several previously unobserved cellular behaviors, e.g., the in vivo rates of protein–DNA associations and an inverse relationship between the duration of DNA replication initiation and replication. Not too much! However, it is not the final word; presently, the mycoplasma model is available as a research tool for the further exploration of parasitic prokaryotes.

### 6.2. Escherichia coli K-12

The same research team led by Markus Covert at Stanford University developed a second WC model, this time for *E. coli* MG1655, the best-studied prokaryote with a ~10 times bigger genome [50,130]. The model also had a modular structure and underwent further improvements.

First, the mathematical structure of the model was unified by merging the assorted modules into the unified gigantic ODE set, probably the largest set in the history of science: 1500 ODEs for the M-matrix and 9000 ODEs for the E-matrix, and 4500 + 4500 equations for RNA and proteins, respectively.

Second, the metabolic M-matrix was presented without a relaxing steady-state approximation. Instead, all reactions were supplied with individual kinetic constants (*k_cat_*, *K_m_*, *V^m^*, etc.) found from 12,000 publications and presented as a function of the intracellular concentration of the respective metabolites. To avoid erroneous pool depletion at each step of numerical integration, the WC model employed the multi-objective function by adding a homeostatic objective: to minimize the sum $\sum_{i}\left|1-C_{i}/C_{0,i}\right|,$ where *C_i_* is the *i*th metabolite concentration, and *C*_0,*i*_ is the respective set point.

Third, the metabolic regulation in the M- and E-matrices was introduced through 22 individual transcription factors (TF) controlling 355 genes in one of three regulatory classes: (i) zero-component (TF becomes instantly active when expressed), (ii) one-component (TF are directly activated or inhibited by a small molecule ligand), and (iii) two-component (TF paired with a membrane protein sensing environment, involving condition-dependent phosphorylation).

Fourth, by improving file I/O and writing inner loops, the simulation runtime of the model was decreased from 10 h to 15 min!

The developed model has been introduced as a tool to integrate and cross-evaluate the extensive molecular data obtained for *E. coli* by various research groups over decades; the found inconsistencies (mostly mismatching published kinetic parameters) were suggested to correct through the sensitivity analyses of the WC model by correcting the suspicious coefficient to a value minimizing simulation error. Three specific inconsistencies were discussed: (i) the underestimated transcription probability of RNA polymerase and ribosomal subunits that failed to reproduce the observed SGR of bacteria, (ii) wrong turnover rates (half-life time) of ~15% of cellular proteins estimated in the published sources by the approximate “N-end” rule, and (iii) the most extensive inconsistencies found in the upgraded M-matrix. The reason for the last inconsistency was explained by the fact that enzyme kinetics in vitro (published data on parameters of the isolated enzymes) were different from that in vivo in the overcrowded intracellular space, especially for membrane-bound enzymes. The problem had to be fixed by adjusting the Michaelis–Menten parameters and use of the multiple objection functions mentioned above. The final intriguing inconsistency was a subgenerational transcription: >50% of the transcriptome, including 72 essential genes of *E. coli*, were found to be transcribed less than once per cell cycle. The survival of cells without essential genes remained a mystery for the authors despite being observed experimentally and reproduced by the model.

In the Appendix B, we discuss the last two extreme cases: (i) a too-low enzyme content in cells with an enigmatic phenomenon of less than one copy of enzyme per cell (Section B.3.4), and (ii) a too-high protein content, causing macromolecular overcrowding (Section B.3.5). In the first case, we concluded that rare enzymes are likely to be an artefact of using proteomic surveys of heterogeneous populations of cells. These enzymatic proteins should not necessarily represent the low number of copies per cell; rather, their source in proteomic analyses could be the rare cells of spontaneous mutants coexisting with the WT. The overcrowding has been reproduced by simple kinetic models that can potentially be applied in GEM to mimic a specific overcrowded environment.

### 6.3. Saccharomyces cerevisiae BY4741

The third WC model [51] was constructed by using the same modular approach as two other WC models. Moreover, the acknowledgment informs us that this reconstruction was done using the WC software developed by the Markus Covert lab for mycoplasma and *E. coli*. Sadly, the WC model was not the only part of the publication, being mixed up with a bulky experimental component on metabolomics and genetic manipulation studies. Consequently, the overloaded publication left many modeling specifics without a clear explanation. Hopefully, there will be other reports using this model and disclosing its features. Meanwhile, we assume that WC models for *E. coli* and *S. cerevisiae* are closely related.

### 6.4. Limitations of the WC Models

Functionally, the largest *E. coli* WC model covers 1214 genes, 28% from the total genome and 45% from the annotated genome. It was lower as compared with the published ME models [1,49,131], and the only apparent progress of the WC models was the upgraded dynamic state for the M-matrix; the positive impact of this upgrade on the model’s performance remains to be proven in future. The skipped genes/processes include: (i) proteases eliminating redundant proteins, (ii) chaperones refolding damaged proteins, (iii) the sigma factors, and a few other factors controlling the transcription. All three groups of macromolecules are crucially important for the self-regulatory behavior of bacteria under nonoptimal growth conditions, especially under stresses. Without them, the model reproduces a defective phenotype of cells able to grow under optimal conditions but expected to be super-sensitive to the smallest deviations from the growth optimum. Thus, the first limitation of the model is the overlooked stress resistance in the reconstructed phenotype.

The indicated deficiency hardly could be revealed in the undertaken study because of the second limitation, neglecting the environmental factors. Even though the growth of bacteria was simulated on three media (minimal and complex aerobic and minimal anoxic), it was not enough to claim that the study ‘explored the effect of growth conditions’. The model never addressed the issue of nutrient limitations or starvation stress or any other stresses. Moreover, the SGR (or corresponding doubling time) was a fixed value throughout the entire simulation.

The third deficiency of the study was the minimal, if any, attempt to verify the model against experimental data by testing the model’s ability to reproduce distinctive behavioral patterns, e.g., the shapes of the growth curves, respiration and fermentation dynamics, proteome profile vs. SGR, etc. Without rigorous experimental verification, the WC model cannot be trusted for its claimed goal to be a tool for the quality control of molecular data.

The fourth limitation is the overstated attention to the cell cycle events, ignoring the other important behaviors of cells (see below).

### 6.5. Cell Cycle as a Part of Growth Kinetics

Cell division cycle is an important part of bacteriology using specialized techniques (time-lapse live-cell microscopy, microfluidics, cells sizing, fluorescent probes, etc.) and dedicated mathematical tools operating with discrete and continuous variables, partial differential equations (PDE), accounting for changes across the time and cell ages [132] and statistics of the frequency distributions of the cells aimed at the discrimination of alternative division mechanisms [133]. It would be wrong to think that a single-cell study is literally limited to one cell; more appropriate is to define it as a bottom-up approach that starts from single-cell observations followed by their assembling into a holistic picture of a heterogeneous population. The source of heterogeneity is the combined effect of numerous factors, including the passing through division cycles and intrinsic stochastic noise. 

The opposite top-down approach operates with bulk population measurements over many cells, followed by inferring the average cell status by using mostly deterministic mathematical models. It is faster, easier, allowing more comprehensive and accurate molecular assays (for comparison, the single-cell proteomics reveals only a fraction of the global proteome assessed by the bulk analyses). The top-down mainstream in microbiology has always been accompanied by bottom-up single-cell studies, which recently have attracted more attention due to remarkable opportunities provided by microfluidics [134,135,136,137,138]. We need both types of research and realize the inherent limitations of each one: the bottom-up single-cell approach focuses on cell division, missing other cellular processes, while the top-down approach neglects cellular heterogeneity and can overlook important mechanisms involving the ‘division of labor’ between individual cells. Most GEMs are deterministic models using ODE and continuous variables (rather than PDE and discrete variables) and, therefore, are better aligned with the top-down data format. The standard output of such GEM includes dynamic data on the asynchronous population of cells evenly distributed across the division cycle, but it can easily be reconfigured into a synchronized format [22] to simulate cell cycle events controlled by the limited number of genes (DnaA—DnaC, FtsZ, septation proteins, etc.). The opposite is not true, and reconfiguration of the specialized cell cycle GEM into the broad-spectrum style is likely to fail because of its narrow scope. Based on this explanation, the only focus of the WC model on the cell cycle seems to be self-restrictive: the whole-cell model should be more inclusive than the single-cell cycle model.

## 7. Conclusions: Strengths, Weaknesses, and Prospects of Specific GEMs

Let us summarize. The undertaken critical review was aimed to reveal the pros and cons of the existing GEMs to identify the most promising directions to move in. We started by defining the terms of simple and complex biodynamics and introduced several basic principles of microbial biokinetics that have not yet been commonly accepted. These principles were summarized as three required conditions to be met by any GEM intending to reproduce the complex microbial biodynamics:(1)to account for the effects of environmental factors on cell dynamics, especially related to the nutrient supply (limiting substrate, its concentrations, and regime of delivery).(2)to simulate the adaptive MMCC changes as conditional gene/protein expressions in response to a changeable environment.(3)to predict the SGR as a complex function of the MMCC status and environmental conditions rather than a fixed input parameter.

**The basic FBA model** meets none of the three conditions, and even a simple growth is presented ambiguously. However, the FBA is still the most popular GEM that gained enviable recognition among diverse users. The FBA is really very good in static applications (e.g., gene knockouts and essentiality prediction), and it is attractive because of the elegant computational resolution of the entire M-matrix. This quality should be preserved but how do we improve the dynamic strength of the FBA without making it too cumbersome? 

A feasible solution could be to couple the basic FBA with the SCM. The SCM itself meets all three conditions above, including a coarse-grained nongenomic reproduction of the MMCC profiles (Figure 3D and Figure 5B). Easy computational steps are required to transform the MMCC profile into the self-adjustable biomass reaction, replacing the fixed reaction (Equation (13)). Then, the stoichiometric coefficients *k_i_s_i_* on the left side and the biomass formulae on the right side can be presented as functions of the *r*-variable (Section B.3.3).

The adjustable biomass reaction links the SCM to the steady-state FBA, making two models interactive. The expected outcome is that every FBA-generated fluxome would become associated with a specific growth pattern. On the other side, any step in the complex biodynamics can be supplemented by a corresponding fluxome screenshot. Each screenshot is static, but their time series produces something like a series of frames in a Walt Disney animation. The range of accessible conditions is practically unlimited due to the flexibility of SCM: any cultivation systems (chemostat, perfusion, fed-batch, and the entire batch process from the lag to death phases); variable media composition; single or multiple limitations; optimal and suboptimal pH; temperature; and other physicochemical conditions, including physiological stresses. Probably, it would be possible to predict the fluxome of mutants vs. the WT under optimal and any challenging growth conditions important for industrial setups. Especially beneficial it would be for the bioprocessing of antibiotics and other secondary metabolites produced only under suboptimal growth conditions.

**The dynamic FBA** models containing the Monod-style exchange module and the fixed biomass reaction do not meet requirements (ii) and (iii). Therefore, they adequately reproduce only simple growth that is of limited use in bioindustry and biomedical applications. A feasible solution could be replacing the outdated Monod model with SCM and upgrading the biomass reaction, as discussed above. The only difference from the FBA–SCM hybrid is that the SCM must be fully integrated into the dFBA, and the biomass reactions need to be updated at each integration step using the SCM *r*-variable as the input. Then, the dFBA–SCM hybrid will be able to simulate the steady-state and transient bioprocesses under diverse conditions with partial characterization of the expression profiles. The expected advantage of the dFBA–SCM hybrid over the state-of-the-art ME and WC models is a relative simplicity and robustness, making it accessible to unexperienced users.

**The published ME models** comply with all three requirements but incompletely: (i) the effect of environmental factors has been incorporated in a ‘relaxed’ way, e.g., by setting constraints on the *s*-variation rather than explicitly by using kinetic equations (Table A1); (ii) the adaptive MMCC changes were simulated only for rRNA and proteins, neglecting other variable cell constituents, e.g., glycogen, polyphosphate, etc.; (iii) the protein expression was simulated using LP and multiple objective functions (maximizing the SGR and protein yield and minimizing the biosynthetic cost) without a clear relationship with the environmental factors; and (iv) the SGR has been both the final output of reconstruction and the factor affecting numerous intermediary steps (e.g., the transcription and translation reactions). Due to the listed omissions, the ME models were not fully successful in a simulation of the complex biodynamics. We showed (Section B.3.2 on the conditional expression of transporters) that introducing a limiting substrate concentration as the independent environmental factor would fix the problem. Other prospects are discussed below, jointly with the WC models.

**WC models.** None of the three required conditions are met in these models [50,130] because of the fixed status of the SGR, MMCC, and growth conditions. Most likely, it was done on purpose to avoid nuisances of a fluctuating environment and stay focused on the molecular complexity. However, the disadvantage of the narrow model’s scope is that its experimental verification cannot be performed completely; the explanation is given below.

**GEM verification against a simple growth can never be complete**. Many modelers prefer simple growth data, assuming that only these data are accurate and reproducible. This is not true; looking back (Figure 1, Figure 2, Figure 3, Figure 4 and Figure 5), we can testify roughly the same experimental scatter for simple growths and complex biodynamics. All depend on the selected experimental techniques, for instance, a fully controlled bioreactor allows recordings of the most challenging unbalanced transient growths with higher accuracy and precision than the simplest exponential growth in shaking flasks. Another misconception comes from a pearl of wisdom: learn to walk before you run. It assumes a step-by-step evolution from a basic model handling simple growth to more advanced versions by adding new modules able to cope with complex biodynamics. However, such a strategy usually badly fails, because the complex biodynamics cannot be decomposed into the series of simple growth steps. It is like trying to learn riding a bike starting from a ‘simplified’ one wheel machine and then step-by-step adding the second wheel, frame, handlebar, and other parts.

WC models are the largest reconstruction in modern microbiology, dealing with a gigantic set of 10,500 mostly nonlinear ODEs without simplifying the steady-state assumptions. This goliath has been applied to the exponential growth between two consecutive divisions, the simplest possible dynamic process. The most advanced part of the model is the combined M- and E-matrices representing TF-regulated metabolic reactions. Some of them are involved in cell division; therefore, the omics data obtained in the synchronized batch culture can be used for experimental verification. However, the rest of the self-regulatory machinery responds to a changeable environment and, therefore, will be not observed in a simple growth experiment! Only the complex biodynamic data carry sufficient information to compare the WC model prediction with observation. It has already been shown in enzymology: non-steady-state (complex) kinetics is much more informative than steady-state kinetics (simple) for the elucidation of catalytic mechanisms [24]. Finally, complex biodynamics is worth studying because of its practical significance; none of the known industrial, in situ, and in vivo processes follow a simple growth pattern.

**Important macromolecules disregarded by ME and WC models.** The SCM splits conditionally expressed macromolecules into P and U clusters (Table 2), supporting the primary and secondary metabolisms responsible for the growth and stress resistance of cells, respectively [22]. These two clusters are revealed in a chemostat at a series of *D* (Figure 5A) or in specially designed titrated batch cultures [139]. The downregulated at high SGR proteins were erroneously called ‘unnecessary’ and even self-inhibitory [140] and then correctly identified as stress response constituents [141,142]. WC and ME models categorized them as unmodeled proteins to be an inert part of the cell mass [123]. It was an unfortunate omission, because the conditionally expressed ‘in inverse order’ (upregulated at a low SGR) macromolecules are crucially important from the system biology and evolutionary perspectives. The microbial growth rate alone is meaningless measure of the fitness without parallel characterization of stress resistance and self-defense against numerous hazardous factors.

**Revisiting the optimality principle**. The philosophy of life is to keep the balance between two general trends: (i) to reproduce themselves as fast as possible under favorable conditions and (ii) to minimize mortality under stress conditions. Cellular resources are limited; therefore, intensive growth and high stress resistance cannot be expressed simultaneously. GEMs aimed at the simulation of a wide spectrum of microbial phenotypes should radically revise the application of the optimality principle. LP in combination with a single objective function such as SGR maximization is appropriate only for nutrient-sufficient optimal conditions. Under restrictive conditions (substrate limitation, nonoptimal temperature, a gradient of inhibitory compounds, etc.), both growth-promoting and stress-resistance subsets of macromolecules should be expressed in proportions that depend on the restriction degree. The older strategy-seeking SGR maximizing under nonoptimal conditions is at risk to produce a degenerative phenotype, a microbial equivalent of immuno-compromised mammals.

## Figures and Tables

**Figure 1 microorganisms-09-02352-f001:**
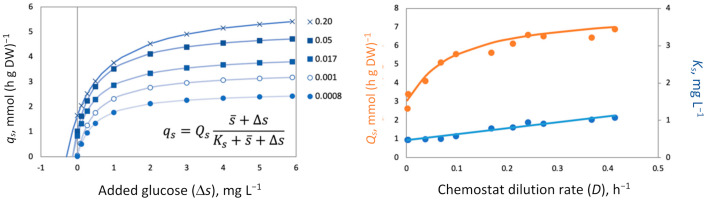
Failure of the Monod model to describe growth with variable MMCC. Yeasts *Schwanniomyces vanrijiae* were grown in a glucose-limited chemostat. **Left:** Short-term response of cells to glucose spike, Δ*s*. The curves are fit to the modified Michaelis–Menten equation, where $\bar{s}$ is the steady-state glucose concentration in the chemostat. The legend shows the *D* and h^−1^. **Right:** The pooled data on *K_s_* and *Q_s_.* Reprinted with permission from [22]. Copyright (1995) Springer Netherlands.

**Figure 2 microorganisms-09-02352-f002:**
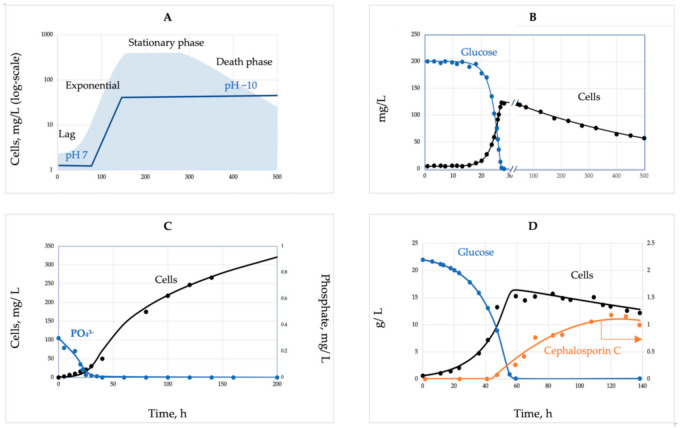
Examples of complex dynamic patterns observed in a batch culture. (**A**) A canonical presentation of batch growth on nutrient broth without a pH control originally deduced by [53]. (**B**) The growth curve of yeasts *Schwanniomyces vanrijiae* on the minimal glucose–mineral medium in a bioreactor [59]. (**C**) Phosphate-limited batch growth of microalga S*elenastrum capricornutum*, under continuous light [59]. (**D**) Growth and antibiotics production in the homogeneous batch culture of filamentous fungus *Cephalosporium acremonium* [60]. The B-D plots fitted to the SCM are reproduced from [59] with permission, Copyright (2019) Elsevier.

**Figure 3 microorganisms-09-02352-f003:**
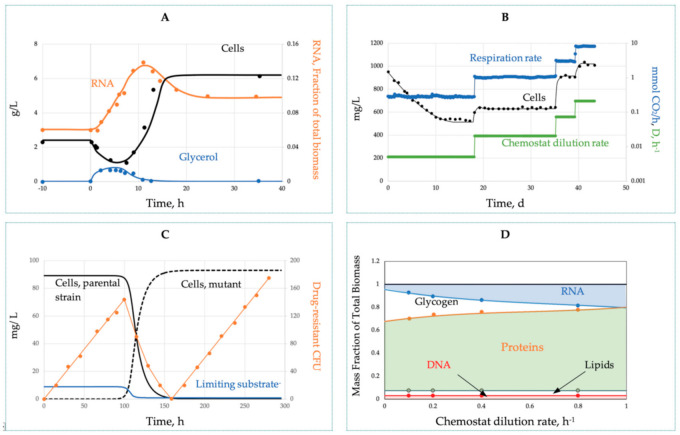
Unbalanced and non-steady-state growth in a continuous culture. (**A**) The shift-up experiment in a chemostat culture of *Aerobacter aerogenes* limited by glycerol; at time zero, the dilution rate was changed from 0.004 to 0.24 h^−1^ [62]. (**B**) Time series of the cell biomass and respiration (CO_2_ formation rate) in a chemostat culture of *E. coli* K-12 after sequential changes of the dilution rates (Panikov, unpublished). (**C**) Spontaneous mutation and auto-selection in a chemostat. At t = 0, one out of ~10^11^ cells mL^−1^ underwent a beneficial mutation improving the substrate affinity (*K_s_* declined), followed by a complete replacement of the parental strain. A genetic sweep was manifested by a decline of the substrate concentration and resetting of the linear accumulation of the drug-resistant cells (Panikov, unpublished rifampicin-resistant CFU data for *Mycobacterium smegmatis*). (**D**) Changes in the cell composition of *A. aerogenes* grown in a NH4^+^-limited chemostat culture as dependent on the dilution rate in a chemostat [62]. All curves were calculated using SCM. Plots A and D are reprinted with permission from [59]. Copyright (2019) Elsevier.

**Figure 4 microorganisms-09-02352-f004:**
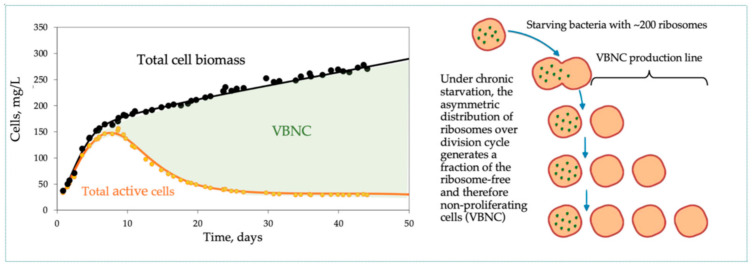
Near-zero growth of Pseudomonas putida F1 in a continuous dialysis culture [28]. **Left**: Growth dynamics and the formation of VBNC (green shaded area). **Right**: The hypothetical mechanism explaining VBNC formation. Similar dynamic patterns were observed in a continuous dialysis culture and in a chemostat with cell recycling. Reprinted with permission from [28]. Copyright (2015) the Society for Applied Microbiology, London, UK.

**Figure 5 microorganisms-09-02352-f005:**
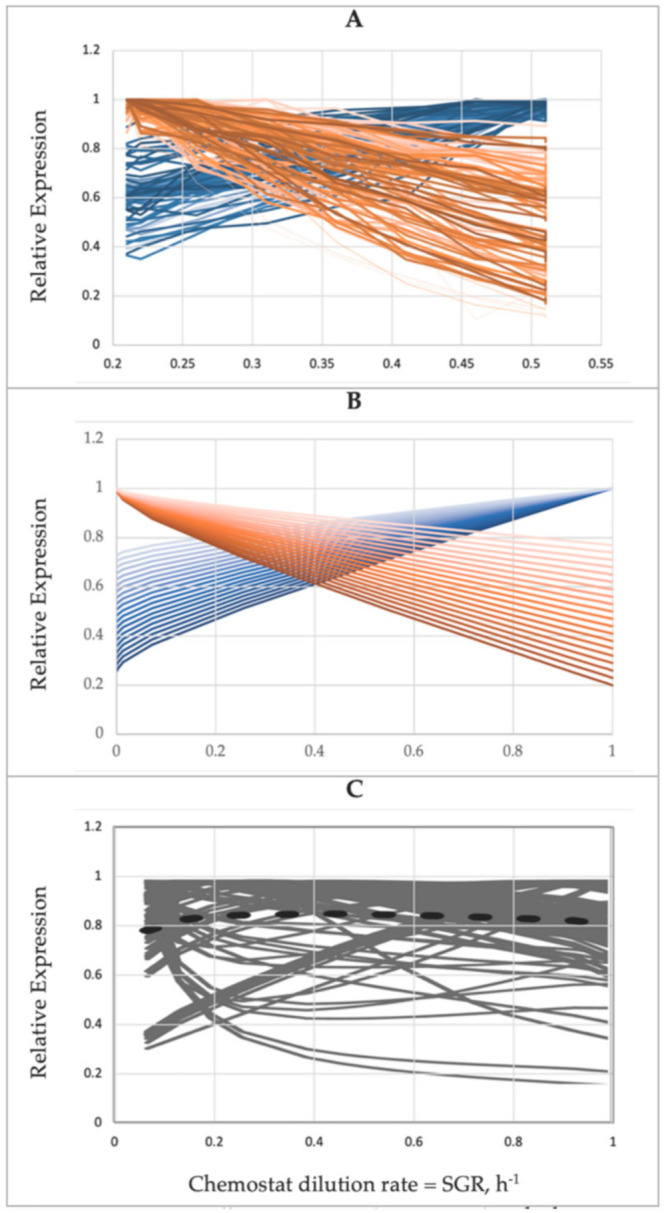
The conditional expression of proteins as dependent on the dilution rate in a glucose-limited chemostat culture of *E coli.* (**A**) Proteomic data [21]. Out of 1525 detected proteins, we selected the most abundant 350 proteins representing 91% of the proteome mass and normalized each protein to its maximum. Blue and orange curves stand, respectively, for the up- and downregulated proteins. (**B**) SCM computing of 40 ad hoc up- and downregulated proteins. (**C**) Profile simulated by the ME model [49].

**Figure 6 microorganisms-09-02352-f006:**
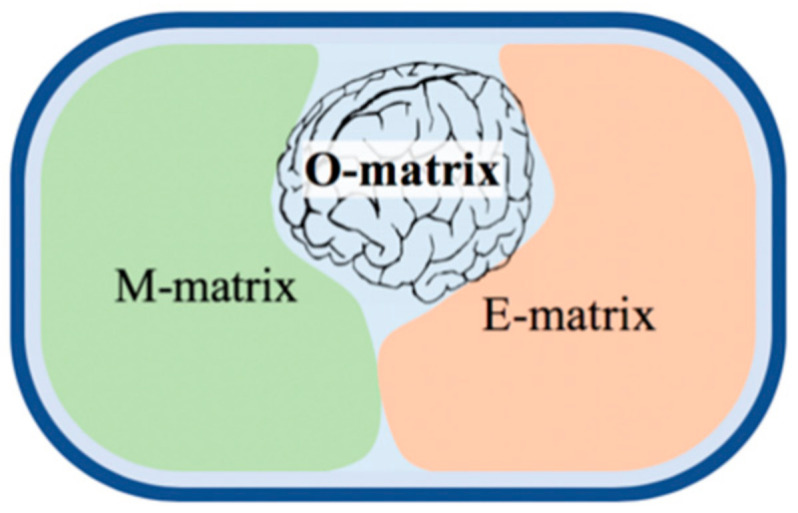
Simplified view on the bacterial genome structure through the prism of GEM modeling.

**Figure 7 microorganisms-09-02352-f007:**
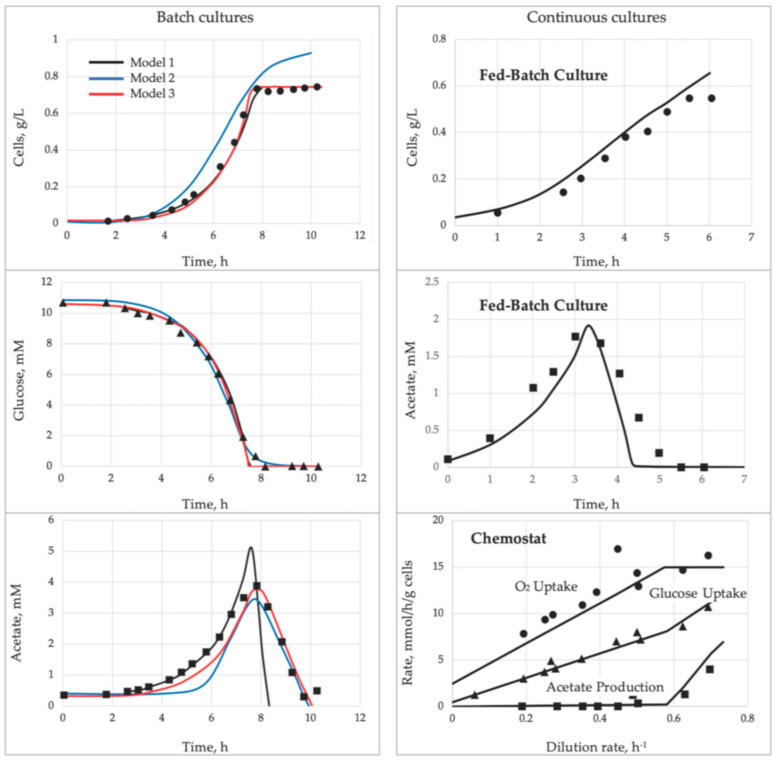
Dynamic FBA simulation of *E. coli* grown in a simple batch culture (**left column**) and in continuous systems (**right column**). The experimental data [86,87] were fitted to the following dynamic FBA models: **Model 1**, the SOA computational approach [87]; **Model 2**, the DOA approach [105]; and **Model 3**, the integrated hybrid FBA/ODE simulation with added transcriptional control [94]. Reprinted with permission from [87], copyright (1994) American Society for Microbiology and with permission from [59], copyright (2019) Elsevier.

**Table 2 microorganisms-09-02352-t002:** Conditionally expressed cell constituents.

P-Components	U-Components
Related to the primary metabolism, required for intensive growth	Related to the secondary metabolism, improve stress resistance
Upregulated under optimal growth conditions	Downregulated under optimal conditions
Ribosomes (rRNA and r-proteins), enzymes involved in translation, and other key cellular processes	Protective pigments, C-storage (glycogen and PHB), antioxidants, high-affinity transporters, enzymes of drug resistance, and antibiotics formation

**Table 3 microorganisms-09-02352-t003:** Deciphering mechanisms controlling the unbalanced and non-steady-state growth [22].

Phenomena	Traditional Interpretation	SCM-Based Clarification
Lag-phase (τ) in batch culture	Metabolic adjustment (poorly predictable)	Lag-phase is predicted based on *r*-state of inoculum (*r*_0_); the higher *r*_0_ the shorter is τ.
The stationary phase in batch culture limited by the C-source	The balance between true growth and cell death	There is no steady-state, *dx*/*dt* < 0, but starving cells differentiate into the active and dormant (VBNC) subpopulations. The cryptic growth is accompanied by upregulation of the U-components improving survival
The death phase	Death rate exceeds growth	
Batch growth limited by the anabolic (conserved) substrates	No explanations to the phenomenon of growth without uptake of deficient element from medium	The total amount of deficient element in the culture remains constant but σ*_P_* progressively declines over time being shared between mother and daughter cells
Shift-up and shift-down in chemostat culture (rise or drop of *D*)	The observed overshoots and undershoots are vaguely attributed to the biological inertia. Prediction is not available	The SCM reproduces inertia by allowing cells to reconfigure their MMCC (proteome profile, ribosomes number). Time delay is automatically generated by Equation (A24) since the pace of *r* change depends on SGR
VBNC formation	A hypothetical ontogenetic stage in the natural life cycle of some bacteria	Ribosome-free cells produced under chronic starvation because of asymmetric distribution of rare ribosomes between mother and daughter cells
Cells response to starvation, the impact of maintenance	Cells dye when the energy supply is below the *m*-level. The maintenance coefficient is constant	Cells survive under deep energy source limitation by adaptive reducing maintenance requirements. Growth becomes slow but never stops
Inverse relationship between SGR and stress resistance	Reasons unknown, interpreted as a descriptive knowledge	Follows immediately from the P- and U-components definition and conservation condition P + U = 1

## Data Availability

Not applicable.

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
