# Peer review of "Genome-Scale Reconstruction of Microbial Dynamic Phenotype: Successes and Challenges"

_microorganisms, 2021, doi:10.3390/microorganisms9112352_

Round 1

Reviewer 1 Report

Dear Authors, the paper "Genome-Scale Reconstruction of Microbial Dynamic Phenotype: Successes and Challenges" fits the scope of the Journal. It is a very comprehensive and significant review, however, in some areas, it fails to review the topic critically. Therefore, some comments should be addressed.

The abstract section is missing information about the niche for this paper, as it should be a contribution to the field, the Authors should explain why is the approach used by them different from other reviews on this topic (i.e. https://doi.org/10.1016/S0167-7799(03)00030-1.

The introduction is short but comprehensive, significant citations are given and an accurate background is presented. The introduction does not define a niche and fails to review the literature critically. Please consider mentioning the need to carry the review and explain how it meets the criterion to be a contribution to the field.

For lines, 83-98 citations should be provided. The information is not originally developed by the Authors.

2.1. Please inform on the opinion of the authors on the development of the approach in the cited literature. Why is MMCC reviewed by the Authors?

Please explain what are the limitations in applying the Michaelis-Menten equation to describe the nutrients uptake. How is this dealt with in literature? How would the authors suggest monitoring cell growth kinetics based on results reported in the literature?

As published GEMs do not deal with explicitly stated growth conditions how would you reconstruct specific phenotypes without the knowledge about the genotype interaction with the environment?

Defining independent and dependent variables requires also the definition of the borders of the experiment, the range and statistical parameters, please discuss the most commonly used p-values in the literature.

Cartoon in line 334 should be used as a figure? Is it truly necessary to prove the point with this figure?

Table 3 and the information should be cited.

Please specify, some of the types of synchronous expression of about 100 genes responsible for resistance to various stresses on E.coli. Opinion, that the expression is not synchronous is also present in the literature.

FBA and ME models are discussed properly and complete.

The given conclusions correspond with the manuscript text.

Reviewer 2 Report

I want to thank the author for letting me the possibility to read and study his paper.

The review presented is a critical picture on the attempts to model the phenotype on the basis of nucleotidic sequences and environmental conditions.

I greatly appreciate the different points of view that the author proposed also in light of the versatility of the discussion

Author Response

Response to Reviewer 2 Comments

Your kind words about the review are highly appreciated.

FYI: we decided to condense the text for easier reading. Now the length of the main text has been reduced down to 21 pages and the Appendix down to 16 pages.

Reviewer 3 Report

This ms presents a review on the use of genome-scale models to analyze microbial dynamics and phenotype. Overall I am in agreement with the content of this ms although I consider that some points need to be improved.

Line 72. Place rephrase to something like "the aim of this study is to show..."

Line 81. Please, rewrite the sentence. Remove "presented verbally".

Page 5. I would suggest to give some more attention to temperature (for example). Please, rewrite your lines on temperature. Temperature influences directly and drastically, microbial and enzyme activity and it is a major key factor affecting microbial growth. Low temperatures could limit metabolic and growth rates and high temperatures could enhanced activity. Of course all within the adequate growth temperature range for each microbial species.

Page 6. Central paragraph. The use and potential of retentostats are slightly mentioned. Perhaps, the potential of this methodology should be clarified and highlighted because it is the only experimental possibility to achieve really low growth rates near the limit to maintenance metabolism. This looks essential to achieve a major understanding of microbial behavior under growth-limiting conditions.

Line 288. Correct the sentence.

Page 7, last paragraph. I do not find adequate to include the figure of the horse and its rider. Please, remove.

Lines 398-400. Perhaps, the authors could indicate that the inflexion point of the growth curve is the only point of measurable growth and "stable" metabolic conditions

Page 11. First paragraph. I would consider the difference of limiting growth and starvation. Starvation does not represent a stable condition, rather some changing conditions leading to death or survival of a number of cells in the population. Growth-limiting conditions (which can be obtained experimentally in a chemostat or retentostat) can represent stable metabolic and growth conditions over time.

Line 444. It looks like a line is missing.

Page 12. I cannot find Table 1. The ms starts with Table 2.

Lines 579-585. This section looks too personal. The author might or not be related with Prof. B. Palsson but this should be reduced to just a line.

Page 20. In this page most of the reported material is too bad. Is this so or there is work that can be considered positive or scientifically useful? Please, consider to highlight some of the good points of those studies and also mention, and justify, those negative aspects.

Lines 929-930. Did the investigator in those studies actually changes arbitrarily the kinetic parameters? Please, look for the reason and try to present a more objective perspective. If possible, please, justify your thoughts.

Line 934. Please rewrite the sentence. That of "compact part" lacks meaning or it does not look appropriate.

Line 942. There are many possibilities of generating heterogeneity in cell populations. Please do not oversimplify. Please, consider removing lines 942 and 943.

About macromolecular overcrowding, there are many potential consequences from overcrowding. Please, review your statements and indicate some of those potential consequences for modeling microbial growth and behaviour (both in the main text of the ms and in the appendix). This to be more elaborated and completed that it is in this version of the ms.

The case of rare proteins is not very satisfactorily treated. Please, consider what you understand by rare proteins. Some could be due to methodological issues, some an artifact, some can be low abundant proteins which are not needed at high concentration or copy number, and some other explainations. What I believe, for sure, is that "rare proteins" can not be safely ignored (line 1710). They might have important activities or regulatory functions which should be included in future models as soon as we understand how they work or what functions they perform.

Are techniques like Neural Networks, among others, considered in the models, or at least, in the training process of those models? I miss the inclusion of this type of methods in a review on past and current models above all for they perspective in a near future and their increase use.

Round 2

Reviewer 3 Report

I am pleased with the authors responses even some that we were initially in disagreement because I believe they are acceptable or reasonable choices.